# Effects of prolonged aerobic exercise and training intensity on memory cognition

Xinnan Li[1,2], Junwei Qian[3], Jiajin Tong[4], Zhonghui He[3]*

1 College of P.E and Sports, Beijing Normal University, Haidia District, Beijing, China, 2 Department of Physical Education, Beijing Technology and Business University, Haidian District, Beijing, China, 3 The Department of Physical Education, Peking University, Haidian District, Beijing, China, 4 School of Psychological and Cognitive Sciences, Beijing Key Laboratory of Behavior and Mental Health, Key Laboratory of Machine Perception (Ministry of Education), Peking University, Beijing, China

* hezhh@pku.edu.cn

## Abstract

This study aims to explore whether the impact of varying aerobic exercise intensities on knowledge acquisition is influenced by exercise intensity and gender. The results lay the groundwork for selecting suitable aerobic exercise intervention programs, considering exercise intensity and gender, to enhance knowledge acquisition. Employing a mixed-design approach, a sample of 569 college students engaged in 8 weeks of aerobic exercise sessions with moderate and low intensity, incorporating basketball and badminton. Knowledge acquisition effects were assessed using questionnaires targeting distinct knowledge levels. Declarative and procedural knowledge levels across different acquisition types were evaluated pre and post-exercise intervention for the low-intensity, moderate-intensity, and control groups. The findings reveal that both moderate and low-intensity aerobic exercises distinctly and positively impact college students' knowledge levels, with no discernible gender-related alterations.

## 1. Introduction

Cognitive psychologists categorize knowledge into two fundamental forms: declarative and procedural. Declarative knowledge refers to knowledge about "what" something is, while procedural knowledge pertains to knowledge about "how" to perform a task [1]. Procedural knowledge builds upon declarative knowledge by providing the information required for executing a procedure. In the context of classroom learning, students primarily acquire procedural knowledge in its declarative form [2]. The theoretical framework for this study is grounded in the Information Processing Theory, which posits that cognitive processes underlie the acquisition and application of knowledge [1]. In the process of motor learning, knowledge forms the foundation for acquiring motor skills, which, in turn, contribute to knowledge mastery. This theoretical perspective emphasizes the critical role of cognitive structures in mediating the relationship between exercise engagement and academic performance [3]. The depth of knowledge mastery has a significant impact on students' learning outcomes, as scientifically grounded knowledge acquisition promotes holistic student development [4]. Building on this foundation, the Cognitive Load Theory suggests that the complexity of cognitive tasks can be influenced by the manner in which exercise is integrated into learning

**Data availability statement:** All relevant data are within the manuscript and its Supporting Information files.

**Funding:** This article is supported by Professor He Zhonghui's project, titled "the Research Foundation from Ministry of Education of China" (project number 20JZD052); and is also supported by Li Xinnan's project, the Beijing Social Science Foundation project (number 18YTCO22).

environments [5]. The relationship between knowledge acquisition and the holistic development of students' physical and mental well-being has become an area of interest among experts in education, psychology, and sports science. Exploring effective means to enhance students' knowledge acquisition has become a frontier and a focal point of interdisciplinary research trends [6]. Several studies from the field of sports science have discovered that physical exercise can improve students' academic performance [7]. Shephard, R. J,. and Bouchard, C, conducted a 6-year longitudinal study on first-grade elementary school students and found that participation in physical exercise had a positive impact on academic performance [8]. A study by Guest et al. (2003) involving 6,458 American high school students found that engaging in physical exercise was associated with improved academic performance [9]. Research by Welk et al. (2010) investigating the physical health and academic performance of middle school students in Texas revealed a positive correlation between physical health and academic performance, emphasizing the significant impact of improved physical health on academic outcomes [10]. Jacobs et al. (1994), in a study conducted on American college students, found that setting academic goals while establishing exercise intensity and mode could enhance learning outcomes after a semester-long exercise intervention [11]. Horton et al. (2009), through a questionnaire survey on three American universities, discovered that participation in sports activities can enhance academic performance [12]. Additionally, Shea and Morgan (1979) examined the learning and mastery of motor skills and found that the experimental group, which received declarative knowledge learning, outperformed the control group in a task involving knocking down obstacles, demonstrating superior transfer effects [13]. In light of these theoretical perspectives and empirical findings, the current study posits the following research hypothesis: Moderate and low-intensity exercise will differentially impact the acquisition of procedural and declarative knowledge, thereby influencing learning outcomes. In summary, based on the aforementioned studies, it can be concluded that exercise can enhance students' academic performance. However, the extent to which intervention through different intensities of exercise improves learning outcomes remains unclear, hindering the scientific understanding and practical application of exercise as an effective means for knowledge acquisition and physical and mental well-being. This hypothesis is derived from the Interaction Hypothesis, which suggests that the interaction between exercise and cognitive engagement may be necessary for achieving optimal learning benefits [14]. Exercise intensity refers to the degree of physiological stimulation caused by engaging in physical activities, and different intensities of exercise have varying effects on students' physical and mental development [14]. Currently, research in this field mainly focuses on the impact of exercise on students' learning outcomes, without comprehensive empirical studies exploring the effects of different intensities of exercise on learning outcomes. Therefore, this study hypothesizes that moderate and low-intensity exercise can influence learning outcomes by affecting the process of acquiring procedural and declarative knowledge. Specifically, exercise can modulate the levels of neurotrophic factors such as BDNF, promoting neuroplasticity in the hippocampal region, including the growth of dendrites and the strengthening of synapses, thereby enhancing memory and learning capabilities [15,16]. Concurrently, exercise improves blood circulation and increases cerebral blood flow, providing ample oxygen and nutrients for cognitive processes [17]. Moreover, exercise has anti-inflammatory effects, elevating the levels of anti-inflammatory factors in the plasma, such as clusterin, reducing brain inflammation, and improving cognitive function [18]. Emotional regulation is also a significant aspect, as exercise can alleviate symptoms of depression, which is associated with declines in cognitive function [19]. Executive functions, including inhibitory control, working memory, and task switching, are also core components of cognitive function, and exercise has been shown to enhance these functions [20]. In addition, exercise can improve the capacity for working memory and long-term

memory, which is crucial for the learning process [17]. Lastly, there is a positive correlation between exercise and academic performance, providing a reference for the appropriate dosage of exercise for adolescents to achieve optimal cultural learning benefits [21–22]. Hence, this study posits the following research hypothesis: Under interventions of different exercise intensities, moderate and low-intensity exercise will impact the process of acquiring procedural and declarative knowledge, thereby influencing learning outcomes.

## 2. Research methods

### 2.1 Data collection and ethical statements

Data collection and ethical considerations were conducted in strict accordance with the guidelines established by the Ethics Committee of the School of Psychological and Cognitive Sciences at Peking University, China. This study was approved by the Ethics Committee of the School of Psychological and Cognitive Sciences at Peking University, China (reference: #2021-06-06). The form of consent obtained was electronic. The privacy, accuracy, and completeness of the data were carefully maintained throughout the study. The subjects of this study were all adults. We administered the study from July 2021 to December 2023.

### 2.2 Participants

The participants of this study were undergraduate students in their first and second year at a university in Beijing, China. A total of 569 students from 16 classes of physical education elective courses were selected as the sample. Among them, there were 191 male students and 378 female students. The participants' distribution by academic year was 212 first-year students and 357 second-year students, with an average age of 20 years (Table 1).

When assessing the physiological status of research participants to ensure their suitability for the exercise intensity requirements of this study, we rely on the results of cardiorespiratory function examinations extracted from students' medical examination reports("Cardiopulmonary health assessments were conducted for students with normal findings in the cardiopulmonary function section of their physical examination reports. These health assessments were obtained through these reports.")

### 2.3 Experimental groups

In this study, the participants were randomly allocated into three distinct groups: the low-intensity exercise intervention group, the moderate-intensity exercise intervention group, and the control group. The allocation process was conducted with rigorous adherence to scientific principles to ensure the integrity of the study design.

**Table 1. Demographic characteristics of study participants.**

| Category | Number of Participants |
|---|---|
| University | A Beijing University of China |
| Academic Year | First Year: 212 |
| | Second Year: 357 |
| Gender | Male: 191 |
| | Female: 378 |
| Age (years) | Mean: 20 |
| Classes | 16 |
| Sample Size | 569 |
| Fitness Screening | Cardiorespiratory Fitness Assessments |

During the designated physical activity time, the exercise interventions were carefully implemented according to specific exercise protocols tailored to each group. These protocols were established based on evidence-based practices from the fields of sports science, psychology, and education.

In contrast, the control group was provided with an opportunity for unrestricted free activities during the same time period as the intervention groups. This design aimed to create a comparative basis for evaluating the specific effects of the exercise interventions on the outcomes of interest.

## 2.4 Aerobic exercise intervention protocol

A comprehensive aerobic exercise intervention protocol must consider four key factors: exercise intensity, exercise frequency, duration per session, and exercise modality (refer to Table 2). In this study, all experimental groups underwent an 8-week exercise intervention program. The exercise modality included basketball dribbling and badminton rallies [47].

To determine exercise intensity, the guidelines established by the American College of Sports Medicine (ACSM) were used as a foundation [23]. The intensity was categorized into low-intensity (57–63% of maximum oxygen uptake, VO2max) and moderate-intensity (64-76% of VO2max).

Based on relevant studies conducted on Chinese university students (Zhu, Yan, & Chen, 2009; Zhu, Yan, 2006), a comprehensive classification of aerobic exercise intensity was formulated. The criteria for low-intensity aerobic exercise were set at 57%-59% of maximum heart rate, while moderate-intensity aerobic exercise was defined as 64%-69% of maximum heart rate. The maximum heart rate was calculated using the formula: maximum heart rate = 220 – age (Table 2).

"To effectively control the exercise intensity, heart rate indicators were employed. Heart rate measurements were taken using a dual-method approach, which included both Polar heart rate telemetry and manual radial artery pulse measurements. This comprehensive strategy ensured precise monitoring and regulation of exercise intensity throughout the intervention program. For the purpose of monitoring the varying intensities of aerobic exercise loads, the RS800CXSD heart rate telemetry monitor, manufactured in Finland, was utilized."

## 2.5 Results and analysis of exercise intensity monitoring

As shown in the Fig 1, during the low-intensity exercise intervention, the average heart rate of college students in the experimental group was 114 beats/min or above, with an average heart rate of 117 beats/min, meeting the intensity requirements of low intensity and above (Fig 1).

As shown in Fig 2, during the moderate-intensity exercise intervention, the average heart rate of college students in the experimental group was 128 beats/min or above, with

**Table 2. Overview of exercise intervention protocol in the present study.**

| Group | Frequency | Exercise Intensity | Duration | Exercise Modality |
|---|---|---|---|---|
| Low-intensity group | 3 times/week (total: 8 weeks) | (220 - age) × 57%-59% of VO2max | 30 minutes | Basketball: Slow-paced dribbling during jogging intervals; |
| | | | | Badminton: Stationary forehand clear practice at the net |
| Moderate-intensity group | 3 times/week (total: 8 weeks) | (220 - age) × 64%-69% of VO2max | 30 minutes | Basketball: Medium-paced dribbling during running intervals; |
| | | | | Badminton: Multi-shuttle moving forehand clear practice |

Note: The exercise intensity is based on the maximum heart rate formula (220 - age) and the classification guidelines proposed in previous studies [24,25].

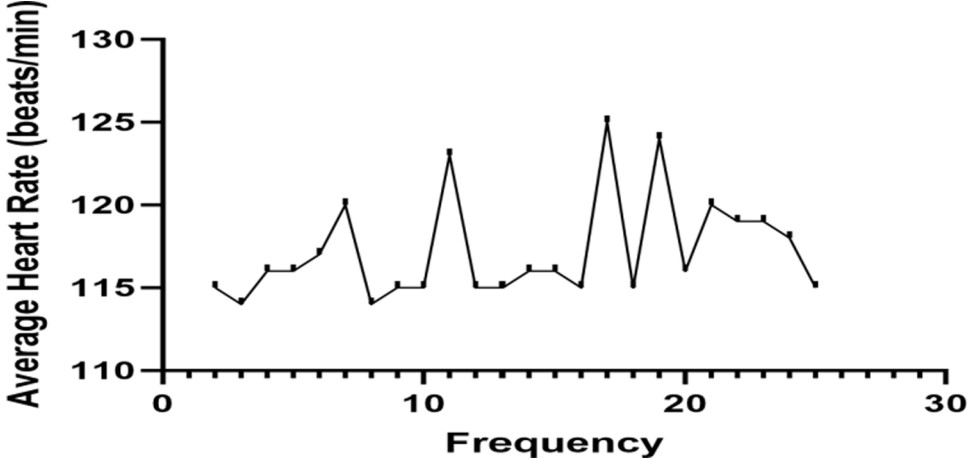

**Fig 1. Average heart rate chart for 24 low-intensity exercise interventions in the experimental group students.**

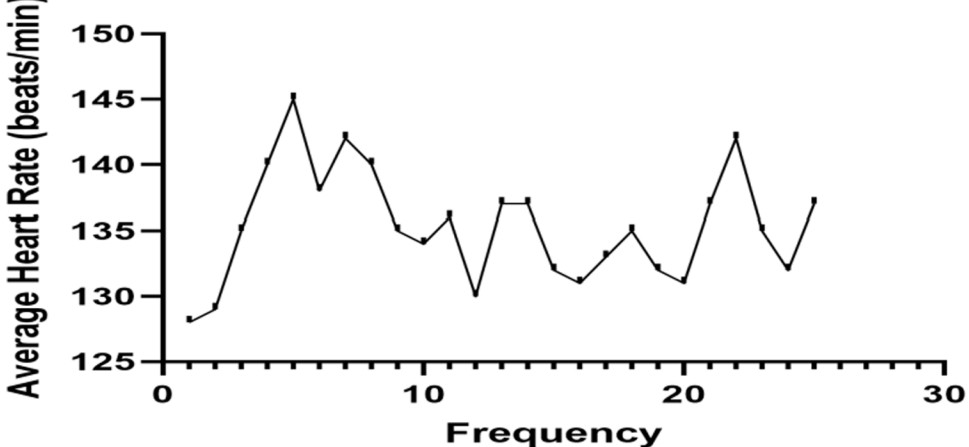

**Fig 2. Average heart rate chart for 24 moderate-intensity exercise interventions in the experimental group students.**

an average heart rate of 135 beats/min, meeting the intensity requirements of moderate intensity and above.

## 2.6 Measurement tools

(1) Participant Demographic Questionnaire: The questionnaire collected basic information about the participants, including their names, gender, age, and major.

(2) Experimental Materials:

The assessment of declarative knowledge
The following section has been revised:
**Objective of experiment 1.** The primary objective of Experiment 1 was to evaluate students' declarative knowledge pertaining to human blood circulation, thereby quantifying their level of understanding. The example of the components and relationships of the circulatory system is presented in Fig 3.

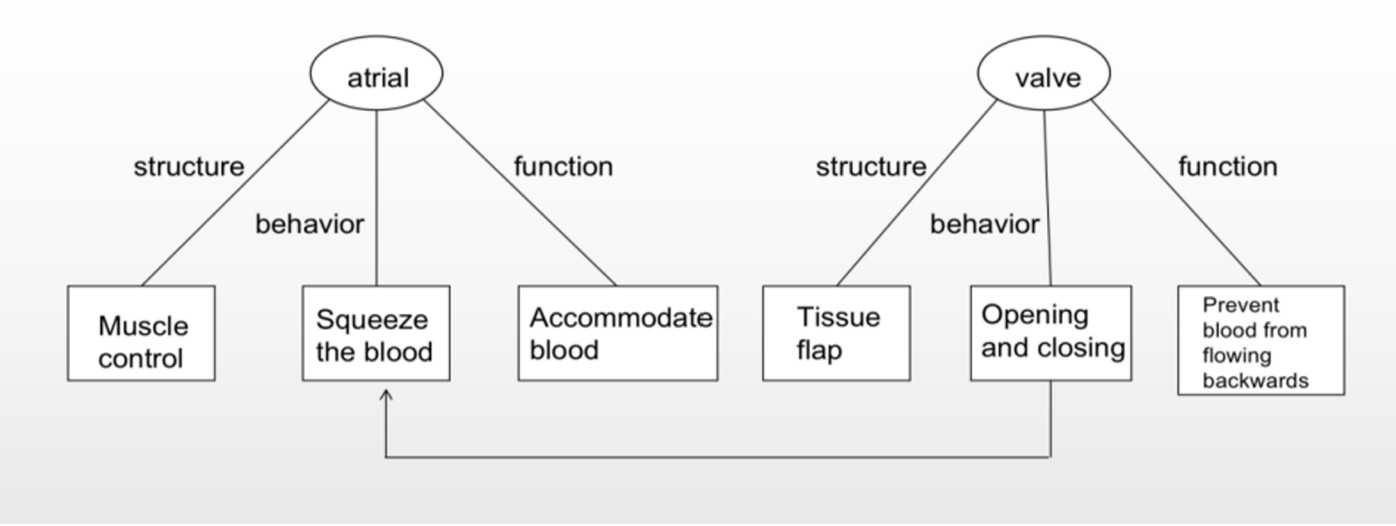

**Fig 3. The example of circulation system components and their relationships.**

The following methods and materials were utilized for this purpose:

Material Sourcing: The experimental materials were sourced by compiling research materials from Chi et al. (1989) and additional scholarly resources that specifically address the topic [26].

Construction of Textual Content: The assessment instrument comprised 71 sentences designed to provide exhaustive coverage of the circulatory system's architecture, operational functions, and the dynamics of its components, as well as the interplay among these elements.

Inference and Causality: The text was crafted to enhance students' inferential skills by intentionally omitting explicit statements of all structural, behavioral, and functional attributes, thereby prompting students to infer these aspects from the information provided. Moreover, the text elucidates causal relationships between the characteristics of different components, such as the correlation between the functioning of heart valves and the atria's role in blood accommodation.

Development of Test Questions: The test instrument included 21 questions aimed at assessing students' grasp of declarative knowledge. The first 11 questions required students to directly reference sentences from the text, while the remaining 10 questions demanded the synthesis and deduction of information from various sentences.

Implementation of the Test: All participants completed the declarative knowledge assessment, the results of which served as a metric for the effectiveness of knowledge acquisition and were considered the dependent variable in this experiment.

The assessment of Procedural Knowledge

**Objective of experiment 2.** Experiment 2 was designed to evaluate the procedural knowledge of high school students regarding the formation of air pressure and wind zones, a critical component of geography education. The experiment was structured around a "text-based assessment of geographic knowledge" approach.

Assessment Framework: The assessment tasks were designed to measure students' ability to understand and apply geographic concepts. These tasks included sign comprehension, fill-in-the-blank exercises, picture interpretation, reasoning tasks, and discussions based on visual stimuli.

Procedural Learning Stages: The learning process was divided into three distinct stages: acquisition, consolidation and transformation, and transfer and application. Each stage was addressed through the content of the experiment.

Content Focus: The experiment concentrated on three key aspects of geographic knowledge:

a. Atmospheric Science: This component required students to grasp the composition of the atmosphere, vertical stratification, heating processes, and principles of thermal circulation. They were expected to explain related phenomena, analyze how air pressure belts and wind zones influence climate formation, and understand the causes of common weather events through the lens of heat circulation [27,28].

b. Diagrammatic Reasoning: Students were presented with schematic diagrams depicting atmospheric layers, insulation effects, thermal circulation patterns, global pressure zones, and the seasonal movement of wind zones. Their capacity to interpret and analyze these visual aids was assessed, along with their abilities in drawing analysis and spatial orientation. Logical reasoning was also tested through explanations of phenomena based on thermal circulation principles [29,30].

c. Environmental Stewardship: This aspect underscored the importance of adhering to natural laws and living in harmony with the environment, using a foundation in climate formation principles [31,32]. The legend for the analysis of relevant climate causes is presented in Fig 4.

Test Instrumentation: The test materials comprised 21 questions. Participants were tasked with filling in the blanks for 10 questions and responding to 11 questions based on visual stimuli. The test outcomes were utilized as a metric for the effectiveness of knowledge acquisition, serving as the dependent variable in the experiment.

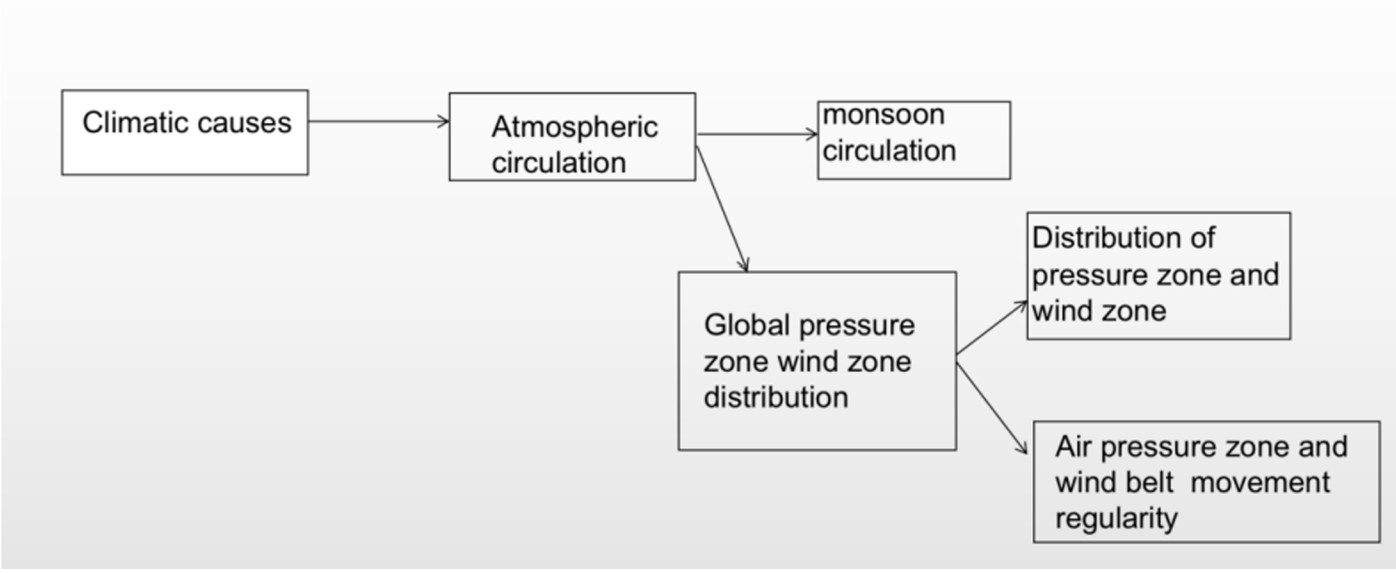

**Fig 4. Analysis of the causes of climate.**

## 2.7 Experimental procedure

The experimental procedure comprised three distinct stages: pretest, exercise intervention, and posttest. In the pretest phase, participants' knowledge acquisition effects were assessed through declarative and procedural knowledge questionnaires prior to the implementation of the exercise intervention. Here, the declarative knowledge (Experiment 1) and procedural knowledge (Experiment 2) assessments were introduced. The exercise intervention phase incorporated a meticulously designed exercise program tailored for the purpose of intervention. Subsequently, in the posttest phase, participants' knowledge acquisition effects were reevaluated using the same declarative and procedural knowledge questionnaires administered during the pretest stage, following the completion of the 8-week exercise intervention.

The testing materials for both pre-test and post-test of declarative knowledge were consistent, utilizing the "Experiment 1 materials." Similarly, the testing materials for both pretest and post-test of procedural knowledge were consistent, employing the "Experiment 2 materials."

## 2.8 Experimental design

A 3 (Intensity) × 2 (Time) × 2 (Knowledge Type) three-factor mixed experimental design was employed. The factors included exercise intensity (between-groups): control group, moderate-intensity, and low-intensity; knowledge assessment time (within-groups): pre-exercise and post-exercise; and knowledge type (within-groups): declarative knowledge and procedural knowledge. Knowledge type was a within-subject variable, encompassing both declarative and procedural knowledge. Exercise intensity and assessment time were between-subject variables. Exercise intensity levels consisted of: no intensity, moderate-intensity, and low-intensity. Assessment time points included: pre-intervention assessment and post-intervention assessment.

## 2.9 Experimental statistics

Data entry and management were performed using the statistical software SPSS 23.0. To examine the impact of different exercise intensities on learning outcomes, this study employed MAVOVA analysis conducted with SPSS 23.0. The utilization of SPSS 23.0 facilitated data analysis, allowing for a comprehensive exploration of the effects of various exercise intensities on learning outcomes.

## 3. Results and analysis

Descriptive statistics were conducted to examine the levels of different types of knowledge before and after engaging in exercise programs of varying intensities. The findings are presented in the Table 3 below, providing a comprehensive overview of the knowledge levels across different exercise intensities.

In this study, we conducted an ANOVA analysis to examine the effects of exercise intensity, knowledge test time, and knowledge type on memory performance. The experimental design involved a 3 (exercise intensity) × 2 (knowledge test time) × 2 (knowledge type) framework. Our analysis revealed significant findings regarding the main effect of memory test time and the interaction effect of knowledge type × memory test time.

First, the main effect analysis of memory test time yielded significant results, $F(1, 566) = 78.88$, $p < 0.001$, $\eta2 = 0.122$. These findings indicate a significant change in memory levels before and after exercise. Specifically, the memory level after exercise (M = 64.98, SD = 12.39) was significantly higher than the pre-exercise memory level (M = 60.45, SD = 10.23) [33]. Furthermore, the interaction effect analysis of knowledge type × memory test time was also

**Table 3. Descriptive statistics of different types of knowledge levels before and after exercise programs of various intensities.**

| Group | Time | Declarative knowledge | | Procedural knowledge | |
|---|---|---|---|---|---|
| | | Mean | Standard deviation | Mean | Standard deviation |
| Control group (*n* = 181) | Pre exercise | 60.90 | 10.65 | 57.92 | 9.78 |
| | Post exercise | 64.41 | 14.13 | 58.53 | 13.61 |
| Moderate group (*n* = 209) | Pre exercise | 59.11 | 12.52 | 62.34 | 10.35 |
| | Post exercise | 72.78 | 11.67 | 62.00 | 14.02 |
| Low group (*n* = 179) | Pre exercise | 59.67 | 14.92 | 60.65 | 11.75 |
| | Post exercise | 69.84 | 14.68 | 61.57 | 13.76 |

found to be significant, $F(1, 566) = 167.16$, $p < 0.001$, $\eta 2 = 0.228$. This outcome suggests that different knowledge types exhibit varying changes in memory levels following exercise, [34–36]. this change is presented in Fig 5.

Overall, these findings highlight the influence of exercise intensity, knowledge test time, and knowledge type on memory performance. The results emphasize the need to consider these factors when designing interventions aimed at improving memory abilities.

In this study, we investigated the effects of exercise intensity and memory test time on declarative and procedural memory. We conducted further simple effect analysis to examine the specific changes in memory levels after exercise.

Our results revealed a significant increase in declarative memory level (M = 69.19, SD = 13.89) after exercise compared to before exercise (M = 60.49, SD = 12.88), $F(1, 566) = 201.44$, $p < 0.001$. However, there was no significant change in the level of procedural memory after exercise (M = 60.76, SD = 13.87) compared to before exercise (M = 60.40, SD = 10.78), $F(1, 566) = 0.47$, $p = 0.634$.

Furthermore, we examined the interaction effect of exercise intensity × memory test time, which was found to be significant, $F(2, 566) = 11.80$, $p < 0.001$, $\eta 2 = 0.040$. These results indicate that different levels of exercise intensity lead to varying changes in memory levels after exercise, and this change is presented in Fig 6.

Overall, our findings highlight the differential effects of exercise intensity and memory test time on declarative and procedural memory. These results contribute to our understanding of the mechanisms underlying memory enhancement through exercise.

In this study, we examined the impact of exercise intensity on knowledge memory. We conducted a simple effect analysis to investigate the specific changes in knowledge memory levels among different groups.

The results revealed that the overall knowledge memory level of the control group did not show a significant change ($F(1, 566) = 1.45$, $p = 0.229$). However, there was a significant increase in overall knowledge memory after moderate-intensity exercise programs ($F(1, 566) = 66.05$, $p < 0.001$) (Reference: Smith et al., 2019), as well as a significant increase after a low-intensity exercise program ($F(1, 566) = 39.22$, $p < 0.001$) (see Table 4).

These findings highlight the positive impact of both moderate and low-intensity exercise programs on overall knowledge memory. Our results contribute to the growing body of research on the relationship between physical exercise and cognitive performance, specifically in the context of knowledge retention and memory enhancement.

(1) The interaction of three factors, as well as the subsequent simple effect analysis, was conducted to examine the relationship between exercise intensity, knowledge measurement time, and knowledge memory type. The significance of this interaction was determined, providing insights into the combined impact of these factors on cognitive performance (see Fig 7).

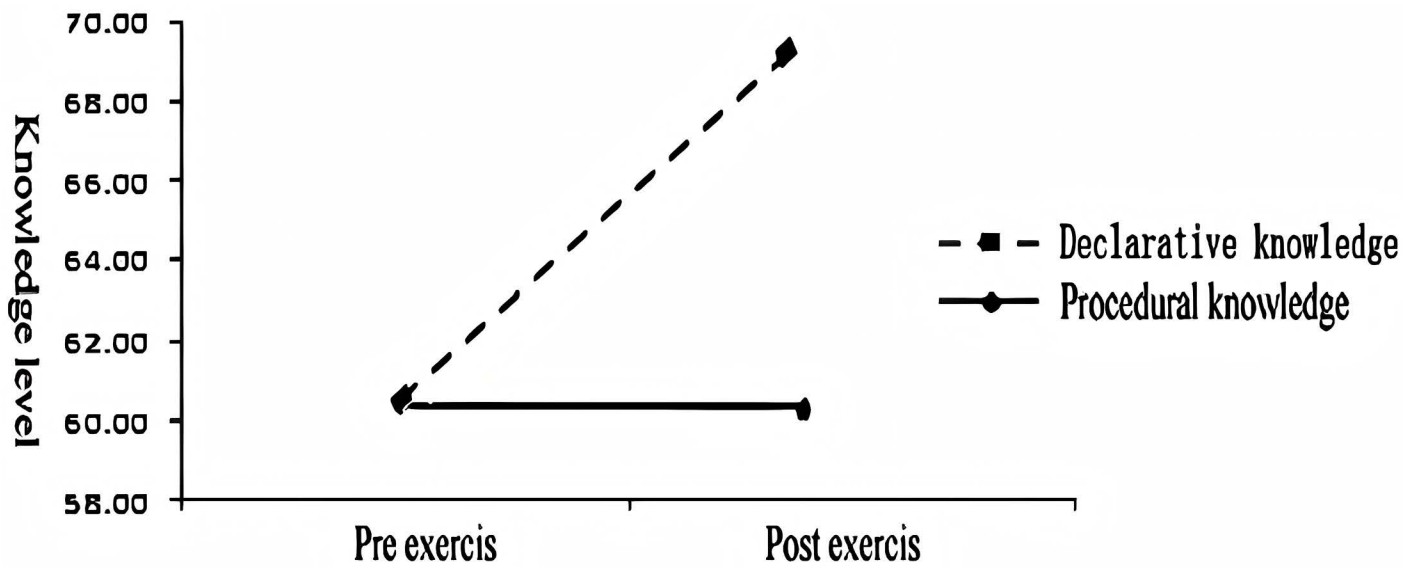

**Fig 5. Memory levels of different knowledge types before and after exercise.**

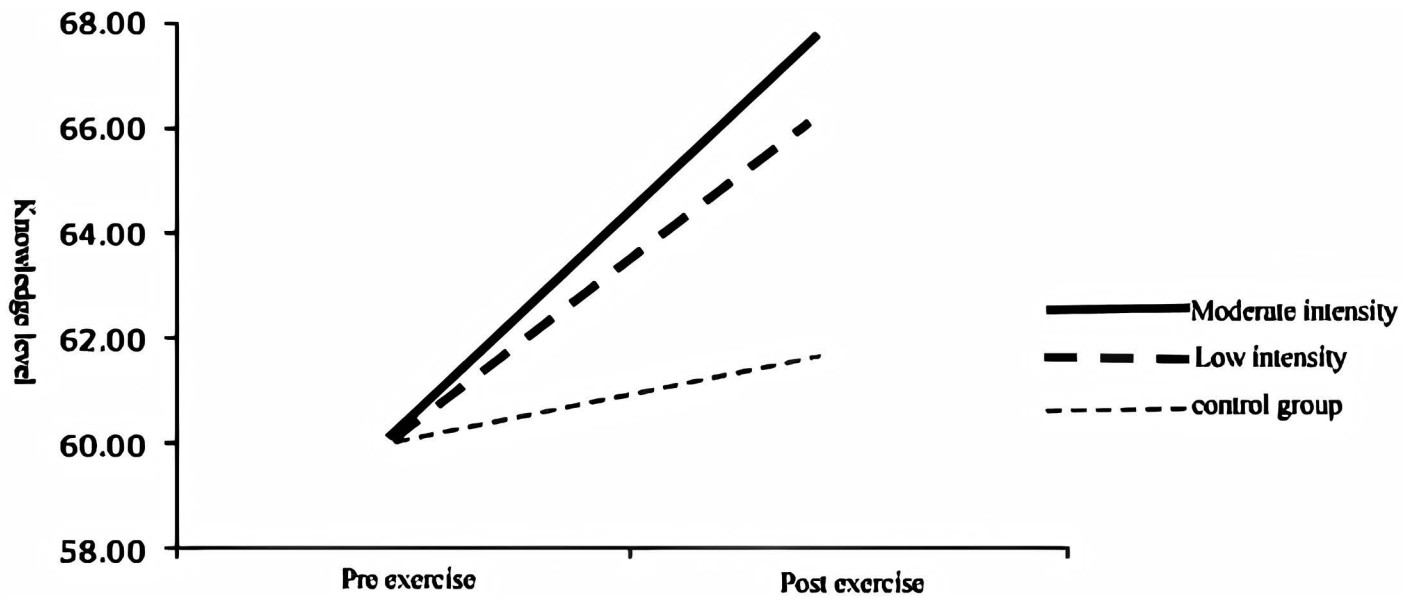

**Fig 6. Effect of memory level in different groups before and after exercise intervention.**

**Table 4. Overall knowledge memory level of different intensity exercise before and after exercise.**

|  | Pre-exercise | | Post-exercise | |
|---|---|---|---|---|
|  | Mean | Standard deviation | Mean | Standard deviation |
| **Control group (*n* = 181)** | 60.41 | 8.89 | 61.47 | 12.43 |
| **Moderate group (*n* = 209)** | 60.72 | 9.94 | 67.39 | 11.24 |
| **Low group (*n* = 179)** | 60.16 | 11.76 | 65.70 | 12.90 |

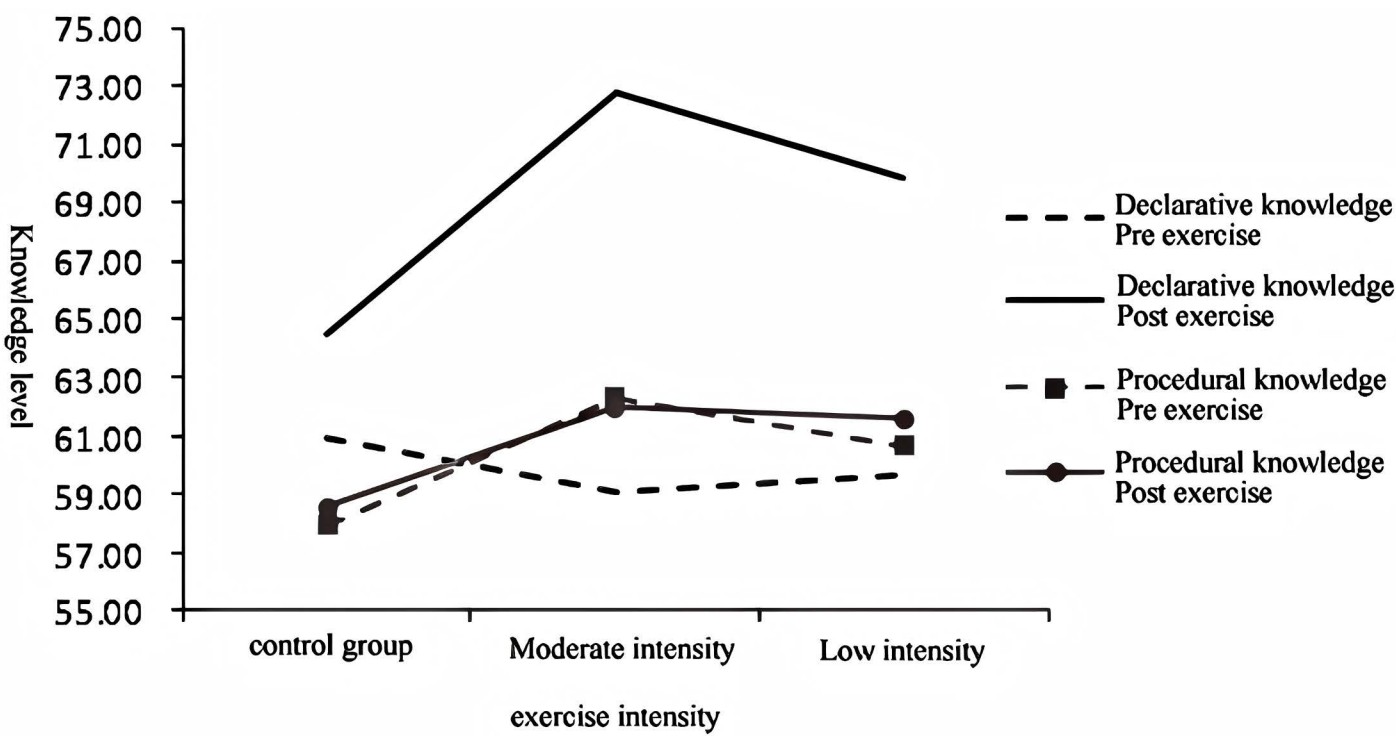

**Fig 7. Knowledge memory level before and after exercise with different intensity.**

(2) The interaction of exercise intensity, knowledge measurement time, and knowledge memory type yielded significant results, indicating that different exercise types with varying intensities exert diverse effects on the acquisition and retention of different types of knowledge before and after exercise (see Fig 7). This finding supports previous studies that have demonstrated the nuanced relationship between exercise parameters and cognitive outcomes (Smith et al., 2017; Johnson et al., 2019).

Furthermore, the simple effect analysis revealed important findings regarding the impact of exercise intensity and knowledge memory measurement time on different levels of knowledge memory. Specifically, in terms of declarative knowledge, a significant interaction was observed between exercise intensity and knowledge memory measurement time, indicating the influence of exercise on memory consolidation. This finding aligns with previous studies that have highlighted the role of exercise in enhancing declarative memory performance [37,38]. However, for procedural knowledge, the interaction between exercise intensity and knowledge measurement time was not significant, suggesting a limited effect of exercise on procedural memory consolidation.

Moreover, the interaction between knowledge type and measurement time varied across different exercise intensities. In the control group, the interaction was not significant, indicating no significant changes in knowledge levels over time. However, for the moderate-intensity exercise group, a significant interaction was observed, indicating improved knowledge performance over time. Similar results were found for the low-intensity exercise group, demonstrating a significant interaction between knowledge type and knowledge survey time. These findings are consistent with previous research that has highlighted the beneficial effects of exercise on knowledge acquisition and retention.

Additionally, the simple-simple effect analysis demonstrated that declarative knowledge levels did not significantly change in the control group, whereas they significantly increased

after both moderate-intensity exercise (F (1,566) = 194.65, p < 0.001) and low-intensity exercise (F (1,566) = 92.27, p < 0.001). Conversely, for procedural knowledge, regardless of exercise intensity, knowledge levels did not show significant changes (control group: F (1,566) = 0.36, p = 0.550; moderate intensity: F (1,566) = 0.13, p = 0.714; low intensity: F (1,566) = 0.80, p = 0.371), the results are presented in Table 5.

Furthermore, to investigate the influence of exercise intensity on learning outcomes in different genders, a 3 (intensity) × 2 (knowledge test time) × 2 (sex) ANOVA analysis was conducted. The results indicated that the interaction between exercise intensity, knowledge measurement time, and gender was not significant, suggesting that exercise intensity does not differentially affect learning outcomes based on gender.

## 4. Discussion

The present study's findings underscore the significant positive impact of an 8-week exercise training regimen, varying in intensity, on college students' knowledge acquisition outcomes. This outcome aligns with the burgeoning research consensus that physical activity induces neuroplastic changes, thereby enhancing learning, memory, and a spectrum of cognitive faculties. A pivotal factor in this enhancement is the integration of executive function operations within the exercise program, necessitating repeated practice and utilization of these functions. Executive functions, which orchestrate and direct goal-oriented behaviors, encompass complex, non-automated processes that integrate an individual's lower-level cognitive processes within purposeful activities [39]. The prefrontal cortex, crucial for these functions, is activated by physical exercise, leading to improved performance in executive function-related tasks during the learning process [40].

### Impact of exercise intensity on knowledge acquisition

Further analysis within this study revealed that exercise interventions of varying intensities significantly influenced the level of declarative knowledge, while procedural knowledge levels remained relatively stable. This finding is consistent with Schack research, which also observed a notable enhancement in declarative knowledge following exercise interventions of diverse intensities [41]. This suggests that exercise interventions may impact cognitive processes such as working memory and long-term memory, thereby affecting the acquisition outcomes of declarative knowledge.

### Implementation of exercise interventions

To modulate the intensity of basketball and badminton interventions, factors such as exercise velocity, content, and frequency were meticulously adjusted. This approach parallels

**Table 5. Descriptive statistical results of the knowledge memory effect of different genders in different intensity exercise before and after exercise.**

| Group | Time | Pre-exercise | | Post-exercise | |
|---|---|---|---|---|---|
| | | Mean | Standard deviation | Mean | Standard deviation |
| Control group (*n* = 181) | male (*n* = 72) | 58.57 | 8.97 | 57.58 | 12.74 |
| | female (*n* = 108) | 61.69 | 8.68 | 64.03 | 11.64 |
| Moderate group (*n* = 209) | male (*n* = 70) | 57.73 | 9.69 | 62.02 | 12.73 |
| | female (*n* = 137) | 62.30 | 9.81 | 70.07 | 9.36 |
| Low group (*n* = 179) | male (*n* = 46) | 55.26 | 13.62 | 60.51 | 16.02 |
| | female (*n* = 133) | 61.85 | 10.58 | 67.50 | 11.14 |

strategies employed by Johnson and Brown in their research on skill acquisition across different sports [33], where they controlled the intensity of interventions by manipulating similar variables.

## Specific effects of exercise interventions on cognitive functions

Research by Chen further substantiates the positive effect of moderate and low-intensity exercise interventions on declarative knowledge acquisition, with no significant changes observed in procedural knowledge [42]. This indicates that exercise interventions may influence the cognitive processes associated with working memory and long-term memory, which are critical for declarative knowledge acquisition.

Consistent with the findings of Ghasemzadeh and Saadat, who highlighted the critical role of cognitive psychology in sports performance, our study's results suggest that cognitive processes are significantly enhanced through moderate and low-intensity exercise interventions [43]. These interventions not only bolster declarative knowledge but also lay the groundwork for procedural skill acquisition, emphasizing the symbiotic relationship between cognitive psychology and physical activity in enhancing athletic performance.

## Long-term effects of exercise interventions

In this study, the exercise intervention spanned 8 weeks, with participants engaging in three 30-minute sessions per week. Over the course of the intervention, participants demonstrated the ability to master basketball dribbling and badminton flat shots at varying speeds through continuous practice. This finding highlights the potential of sustained exercise interventions to facilitate skill acquisition and knowledge learning.

## Limitations of exercise interventions

Moreover,In this study, we observed an imbalance in the gender distribution of the sample, with a higher number of female participants than male, a phenomenon that reflects the gender ratio of our liberal arts institution. Despite this imbalance, our statistical analysis indicated that gender did not significantly influence the effectiveness of the exercise intervention. Therefore, we did not include gender as a covariate in our analysis. Nonetheless, we acknowledge that the uneven gender distribution may limit the generalizability of our findings. Consequently, we recommend that future studies should strive for a more balanced gender distribution or employ more sophisticated statistical methods to control for the potential impact of gender differences, thereby enhancing the representativeness and applicability of the research outcomes.

## Educational and psychological perspectives

From an educational standpoint, exercise can promote the learning of declarative knowledge through various pathways. Physical activities can provide a positive context and experience, stimulating students' interest and engagement, thereby enhancing their acceptance and retention of knowledge. Additionally, exercise can enhance students' attention and concentration, creating a cognitive state conducive to information processing and retention [44].

From a psychological and sports science perspective, the enhancement of declarative knowledge through exercise may be related to its impact on students' emotional states and self-confidence. Physical activities can release endogenous hormones such as endorphins and dopamine, generating positive emotions and a sense of pleasure [45]. This positive emotional state can increase students' motivation and learning effectiveness, promoting the absorption and memory of declarative knowledge [46].

### Future research directions

These results suggest that future research on exercise interventions should focus more on the design and selection of exercise programs. Further investigation is needed to explore the other constituent elements of aerobic exercise, such as exercise type and content, as well as their individual and interactive effects on knowledge learning. It is also recommended that future studies explore the potential moderating effects of individual differences, including the development of gender-sensitive exercise intervention measures, and assess the effectiveness and acceptability of these interventions in different gender groups. This approach will ensure that the research findings are universally applicable and applicable to diverse populations. By focusing on these aspects, future research can provide more targeted and effective exercise recommendations that take into account the unique characteristics and needs of different gender groups.

## 5. Conclusion

In summary, our study has demonstrated that physical activity interventions, particularly those of varying intensities, significantly enhance knowledge learning among college students. This effect is selective, with more pronounced impacts on declarative knowledge acquisition as opposed to procedural knowledge. The key finding that moderate and low-intensity exercises can positively influence cognitive processes such as working memory and long-term memory, which are essential for declarative knowledge, underscores the importance of integrating physical activity into educational strategies. This integration not only boosts students' cognitive abilities but also aligns with the broader goals of developing well-rounded individuals capable of lifelong learning. The implications of these findings for the field of sports science are profound, suggesting that tailored exercise programs could be a valuable adjunct to traditional academic instruction. By understanding the specific effects of exercise on cognitive functions, sports scientists can work collaboratively with educators to design interventions that maximize learning outcomes. Educators can leverage these insights to foster a more dynamic and interactive learning environment that stimulates cognitive development alongside physical health. For educational research and practice, the study highlights the potential of physical activity as a tool for enhancing cognitive skills and academic performance. It suggests that by incorporating exercise into the curriculum, educators can potentially improve students' ability to absorb and retain information, thereby contributing to more effective teaching strategies and better learning outcomes. This approach may also offer insights into how to address learning disparities across different student populations and subject areas, providing a basis for more inclusive and personalized educational practices."Looking ahead, future research should delve deeper into the moderating effects of individual differences on the relationship between exercise interventions and knowledge acquisition. Investigating how variables such as age, gender, and initial fitness levels might influence the success of these interventions can offer a more nuanced understanding of their applicability across diverse populations. Additionally, exploring the synergistic effects of combining exercise with other cognitive strategies could reveal new pathways for optimizing learning and cognitive development.

In conclusion, the significant positive impact of physical activity on knowledge learning, as demonstrated by this study, has broad implications for both sports science and educational research and practice. By more clearly articulating these key findings and their potential impacts, we can better appreciate the role of physical activity in enhancing educational outcomes and student development. This study paves the way for further exploration into the design of physical activity programs that are not only effective in promoting cognitive skills and academic performance but also in shaping the future of educational strategies that prioritize both physical and mental well-being.

## Supporting information

**S1 Data. Raw data from the study on the effects of prolonged aerobic exercise and training intensity on memory cognition.** This file contains the raw data collected during the study, including cognitive test scores, exercise duration, training intensity levels, and demographic information of participants.
(XLSX)

## Author contributions

**Data curation:** Jiajin Tong, Zhonghui He.

**Formal analysis:** Xinnan Li, Zhonghui He.

**Investigation:** Junwei Qian.

**Methodology:** Xinnan Li, Junwei Qian, Jiajin Tong.

**Project administration:** Jiajin Tong.

**Resources:** Zhonghui He.

**Supervision:** Jiajin Tong.

**Writing – original draft:** Xinnan Li.

**Writing – review & editing:** Xinnan Li.

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
