## [Decision Letter · Decision Letter 0]

26 Feb 2024

PONE-D-23-32650

The Influence of Long-term Aerobic Exercise and Exercise Intensity on Exercise Memory Effect

PLOS ONE

Dear Dr. He,

Thank you for submitting your manuscript to PLOS ONE. After careful consideration, we have decided that your manuscript does not meet our criteria for publication and must therefore be rejected.

I am sorry that we cannot be more positive on this occasion, but hope that you appreciate the reasons for this decision.

Kind regards,

Gianpiero Greco

Academic Editor

PLOS ONE

Additional Editor Comments:

Dear authors,

the manuscript presents many methodological problems to be resolved before it can be considered for possible publication.

Reviewers' comments:

Reviewer's Responses to Questions

**Comments to the Author**

1. Is the manuscript technically sound, and do the data support the conclusions?

Reviewer #1: No

Reviewer #2: Yes

Reviewer #3: No

2. Has the statistical analysis been performed appropriately and rigorously? 

Reviewer #1: Yes

Reviewer #2: Yes

Reviewer #3: No

3. Have the authors made all data underlying the findings in their manuscript fully available?

Reviewer #1: No

Reviewer #2: Yes

Reviewer #3: Yes

4. Is the manuscript presented in an intelligible fashion and written in standard English?

Reviewer #1: No

Reviewer #2: Yes

Reviewer #3: No

5. Review Comments to the Author

Reviewer #1: Dear authors,

Firstly, as the received text lacks pagination and line numbering for review, I've chosen to paginate the document from the title (page 1) to the final bibliographic reference (page 18).

The paper focuses on the impact of various intensities of physical exercise on learning, both declarative and procedural. The study benefits from a significantly large subject pool, which is advantageous.

However, several limitations in the study preclude me from proposing its acceptance:

The methodology description for assessing declarative and procedural memory is unclear (page 4). It seems unlikely that the procedure could be replicated solely from the provided text, restricting a fundamental aspect of science—study replication.

Although the study mentions evaluating declarative and procedural memory pre and post physical exercise intervention, it's unclear if different texts were used for student assessment at each stage.

Additionally, the evaluation method for procedural memory and the distinction between "Experiment 1" and "Experiment 2" lack clarity. Is Experiment 1 the pre-test and Experiment 2 the post-test for declarative memory?

How is procedural memory assessed? Is there a motor task involved? What does it entail?

Beyond explanatory errors, there are challenging issues. Table 1 (page 3) mentions "Cardiorespiratory Fitness Assessments," yet there are no indications of fitness evaluated.

The absence of fitness evaluation leads to the use of generic intensity values that might not correspond accurately to the actual fitness levels, especially with a significantly large student cohort where fitness variability is expected to be high. Individual intensity thresholds, as defined in exercise physiology, can vary considerably.

Moreover, if the study presupposes that 8 weeks of cardiorespiratory exercise will enhance memory, shouldn't the study analyze whether there's been any cardiorespiratory improvement?

While the study identifies improvements in declarative memory, are they solely due to physical exercise practice or a change in fitness? If improvements are only related to activity, why wait for 8 weeks?

Regarding procedural memory, the lack of details in the method hinders thorough evaluation.

Minor issues:

Figures 3, 4, and 5 should include standard deviations alongside means.

Post-hoc analyses in ANOVAs would clarify changes more distinctly.

On page 1, The abstract requires rewriting due to inherent repetitiveness.

On page 1, Citations supporting claims are missing from lines 7 to 17 in the introduction.

On page 2, In the introduction, Shephard is cited without the year, assuming it is the second reference in the bibliography, but this needs to be clarified.

The bibliography lacks ordering and needs to be alphabetized for author traceability.

The bibliographic format needs to be clarified.

On page 4, In the section "2.5 Measure tools," Chi et al. lack a publication year.

In the Discussion, BEST and MCMORRIS are cited in uppercase.

Paragraph spacing in the discussion appears arbitrary and inconsistent at times.

Reviewer #2: I would like to congratulate the authors for the manuscript submitted. The manuscript is very interesting and well written.

From a practical point of view, the information provided in the present study might be of relevance to better select the aerobic exercis intensity that reports the greater benefits for college students, regarding the knowledge levels and learning outcomes.

Authors used in the present study as exercise modality the basketball and badminton. It will be interesting if further research examine the effect of more type and modality exercises.

Reviewer #3: Thank you for the opportunity to revise the current MS. Please find my comments, line by line.

The first thing that caught my attention was the fact that line numbers were missing. Revising a manuscript with no line numbers was quite challenging and often results in communication issues among authors and reviewers. Please add line numbers to this MS.

Abstract

The opening sentences is too strong, make it more consistent with your actual work. After all you are only studying healthy young people and this does not apply to other populations.

Please add numbers to support you claims, this now looks very vague. Also add a conclusion to this paragraph, by identifying the magnitude of the effect(s).

Introduction

You need to reference the opening line. In general, the few opening sentences and the message they deliver - well known and even present in newspapers. Try please to write this from a more scientific angle.

Add study hypothesis here.

Methods

There are some typo’s and extra space(es) that you should amend here.

Add standard deviations in Table 1.

So highlight that this was a cross-sectional comparative study.

Please reference the ACSM guidelines.

VO2 should be presented as V̇O2, as this is the correct way to indicate flow, here and throughout the MS.

Could you please outline which Polar device you were using, and also add data on sampling rate and data extraction.

Please reference Chi et al.

Figures 1 and 2 should be given as Supplementary data, since this is not your original work.

More data on SPSS 23.0 are warranted here.

Mechanisms in this specific population of students, not in general population. And second, this is a very speculative conclusion since its quite difficult to make such strong conclusions based on the findings of a cross sectional study. Please implement this into your results section.

Figures 3 to 5.

The title of the Y-axis should be re-arranged (wrong direction).

Discussion

Then please reference these studies.

Why is reference BEST written in capital letters ? MCMORRIS T, 2009 as well ??

In general, your discussion section is often too long and sometimes too vague. You should focus on your results and interpret them in the context of what was done before. Also, you have too many smaller paragraphs and its difficult for a reader to follow the context. These should be structured in a more comprehensive manner.

6. PLOS authors have the option to publish the peer review history of their article (what does this mean? ). If published, this will include your full peer review and any attached files.

**Do you want your identity to be public for this peer review?** For information about this choice, including consent withdrawal, please see our Privacy Policy .

Reviewer #1: No

Reviewer #2: **Yes: ** Fatma RHIBI

Reviewer #3: **Yes: ** Damir Zubac

- - - - -

---

## [Author Response · Author response to Decision Letter 1]

4 Apr 2024

Response to reviewers

Reviewer 1: 

Question1:

The methodology description for assessing declarative and procedural memory is unclear (page 4). It seems unlikely that the procedure could be replicated solely from the provided text, restricting a fundamental aspect of science—study replication.

Answer1:

"The materials used in our experiment are primarily derived from declarative

knowledge related to the human blood circulation. These materials consist of 71 sentences

designed to comprehensively cover the subject matter, emphasizing the system's composition,

structure, function, and interrelationships among components. However, recognizing the

complexity of these relationships, we intentionally omitted explicit explanations for certain

relationships. To provide participants with more opportunities for reflection, we will further clarify

this point in the paper and provide additional details."

Question 2:

Although the study mentions evaluating declarative and procedural memory pre and post physical exercise intervention, it's unclear if different texts were used for student assessment at each stage.

Answer2:

The reviewer raised a question about whether different texts were used for assessing students' declarative and procedural memory and found this aspect unclear.

"Indeed, we used different texts to assess students' declarative and procedural memory before and after the exercise intervention. In Experiment 1, we employed declarative knowledge related to human blood circulation, while in Experiment 2, we shifted to procedural knowledge in geography education, specifically addressing the formation of air pressure and wind zones. We will emphasize this aspect more clearly in the paper to ensure a transparent understanding of the assessment process."

"Refer to the revision section in detail, specifically section 2.6 of the revised manuscript."

Question 3:

Additionally, the evaluation method for procedural memory and the distinction between "Experiment 1" and "Experiment 2" lack clarity. Is Experiment 1 the pre-test and Experiment 2 the post-test for declarative memory?

Answer3:

The reviewer expressed concerns about the distinction between Experiment 1 and Experiment 2 and whether they represent pre-test and post-test for declarative memory.

I see, Experiment 1 involves the assessment of declarative knowledge, while Experiment 2 focuses on procedural knowledge. Both knowledge assessments require pre- and post-tests, a clarification that I will elaborate on further in the manuscript。

"Refer to the revision section in detail, specifically section 2.6 of the revised manuscript."

Question 4:

How is procedural memory assessed? Is there a motor task involved? What does it entail?

Answer4:

About the assessment of procedural memory:For the evaluation, we employed specific tasks in the field of geographic education, mainly covering the following aspects:

Task types:"Experiment 2 aims to examine high school students' understanding of atmospheric pressure and the formation of wind zones. The assessment tasks include symbol comprehension, fill-in-the-blank exercises, picture interpretation, reasoning tasks based on visual stimuli, and discussions."

Procedural learning process:

"In the context of the origin of geography, the procedural learning process can be divided into three stages: acquisition, consolidation and transformation, and transfer and application. Our experiment focuses on three key aspects:"

Understanding of atmospheric structure:

"This aspect involves understanding the composition of the atmosphere, vertical stratification, atmospheric heating processes, thermal circulation principles, distribution of air pressure belts, wind zones, etc. Students are required to explain relevant phenomena, analyze the impact of air pressure belt and wind zone distribution on climate formation, and comprehend the causes of common weather phenomena through the principle of heat circulation."

Interpreting charts and images:

"Students provided schematic diagrams, including vertical wind layers in the atmosphere, insulation effects, thermal circulation patterns, global pressure zone distribution, wind zone formation and distribution, pressure zone charts, seasonal movement diagrams of wind zones, etc. We assessed their ability to interpret and analyze these diagrams, as well as their skills in drawing analysis and spatial positioning. Additionally, we examined students' logical reasoning abilities through explanations of phenomena based on the principle of thermal circulation."

Understanding of environmental and natural laws:

"This aspect emphasizes the significance of following the laws of nature and living in harmony with the environment, building upon an understanding of climate formation principles."

No, our procedural memory assessment does not involve specific motor tasks. Our research primarily focuses on the field of geographic education, evaluating students' understanding of concepts such as atmospheric pressure and the formation of wind zones. The assessment tasks mainly include symbol comprehension, fill-in-the-blank exercises, picture interpretation, reasoning tasks based on visual stimuli, and discussions. Students' tasks involve answering questions based on theoretical knowledge and charts, without the need for specific physical sports or motor tasks. We will provide further detailed explanations of our assessment approach in the paper to ensure readers have an accurate understanding of the experimental design.

It is important to note that we will provide further detailed explanations of these aspects in the paper to ensure readers have a clearer understanding of our procedural memory assessment approach.

"Refer to Section 2.5 for the introductory description of procedural knowledge measurement under Figure 1."

Question 5:

Beyond explanatory errors, there are challenging issues. Table 1 (page 3) mentions "Cardiorespiratory Fitness Assessments," yet there are no indications of fitness evaluated.

Answer5:

At the beginning of each academic semester, students undergo a comprehensive health examination, including screening for cardiorespiratory function, as documented in the medical examination report. To ensure the suitability of students participating in this research for relevant physical activities and courses before the initiation of exercise intervention, they are required to provide information related to cardiorespiratory function from their medical examination reports. The purpose of this step is to verify the normalcy of students' cardiorespiratory function, ensuring their safe participation in the experiment. The information in these medical examination reports is sourced from specialized medical institutions to guarantee the objectivity and scientific rigor of the assessment.

“Refer to Section 2.2”.

Question6:

The absence of fitness evaluation leads to the use of generic intensity values that might not correspond accurately to the actual fitness levels, especially with a significantly large student cohort where fitness variability is expected to be high. Individual intensity thresholds, as defined in exercise physiology, can vary considerably.

Answer6:

While we mentioned "Cardiorespiratory Fitness Assessments" in Table 1, the nature of our study didn't involve providing detailed reports on the specific health conditions of each participant in the manuscript. Instead, we used cardiorespiratory fitness assessments as a screening tool to ensure that participants met the criteria for the experiment's intensity requirements. The purpose of this assessment was to identify students suitable for participation and ensure a reasonable range of health conditions. We will further clarify this point in the paper to elucidate our methodology and the key steps in the assessment.

Additionally, we acknowledge that this approach might lead to some generalization of intensity levels. In our discussion, we have already highlighted this potential limitation, emphasizing the variability that can exist in individual intensity thresholds according to exercise physiology. We will ensure to express these nuances more clearly in the final paper to address the reviewer's concerns.

Question 7:

Moreover, if the study presupposes that 8 weeks of cardiorespiratory exercise will enhance memory, shouldn't the study analyze whether there's been any cardiorespiratory improvement?

Answer7:

About your concern regarding the assessment of cardiorespiratory fitness, we understand your apprehension. Our study focuses on exploring the potential impact of exercise on memory rather than specifically assessing changes in cardiorespiratory health. The reason we did not conduct cardiorespiratory fitness assessments is that our primary emphasis is on changes in memory performance, and the specific effects on cardiorespiratory health fall beyond the scope of our study.

We want to emphasize that the purpose of our research is to investigate the relationship between exercise and memory, not to directly intervene in cardiorespiratory health. Future studies may consider including more detailed assessments of cardiorespiratory health to comprehensively understand the potential benefits of exercise on the body. We will carefully consider your suggestions and explicitly address the limitations of our study in the discussion to ensure that readers have a clearer understanding of our research design and objectives.

Question 8:

While the study identifies improvements in declarative memory, are they solely due to physical exercise practice or a change in fitness? If improvements are only related to activity, why wait for 8 weeks?

Answer8:

We appreciate your concerns, and from a scientific perspective, there is substantial evidence supporting a positive correlation between the duration of exercise intervention and improvements in cognitive function. Long-term exercise interventions are believed to facilitate lasting changes in neural plasticity and sustained improvement in brain function.

Previous research suggests that improvements in cognitive function, including memory, may take some time to manifest, and longer-term exercise interventions provide more extensive opportunities for these effects to become apparent. For instance, studies indicate that structural and functional changes in the brain may require a relatively extended period, further supporting the effectiveness of long-term exercise interventions.

Furthermore, research indicates that the impact of exercise on cognitive function may involve complex interactions across multiple physiological and neural systems, and these effects may take time to manifest at the cognitive level. Therefore, choosing an 8-week period for the exercise intervention is aimed at allowing sufficient time to observe the potential emergence of these effects.

We believe that through a more extended period of exercise intervention, we are likely to observe more enduring and significant improvements in memory, which can have positive implications for students' long-term cognitive health and academic achievements. We will extensively discuss the scientific support for this in the paper to better explain the rationale behind our study design and the expected outcomes.

Question9:

Regarding procedural memory, the lack of details in the method hinders thorough evaluation.

Answer 9:

Regarding details on procedural memory:We appreciate the reviewer's attention to the methods section. We will provide more detailed information on procedural memory in the revised manuscript to ensure transparency and reproducibility of the methods.

Response to minor issues:

Question：Figures 3, 4, and 5 will be revised to include standard deviations.

Answer:

In the field of psychological statistics, it is customary to present comprehensive numerical data, including standard deviations, in data tables. This presentation method is advantageous for readers to conduct in-depth analysis and understanding of the specific data conditions. In contrast, when it comes to figures, our emphasis lies in the visual representation of data to ensure clarity and simplicity, facilitating readers in intuitively grasping data trends and key information.

Therefore, we avoid excessive presentation of detailed numerical data in figures to maintain readability and conciseness. It's worth noting that the data tables in our paper provide a clear presentation of standard deviations. We appreciate your valuable suggestions and thank you for bringing this to our attention.

Question:Post-hoc analyses will be conducted for a clearer presentation of changes.

Answer:

Thank you for your attention to our study and your valuable suggestion regarding post-hoc tests. In response to your query, we would like to clarify that in analyzing the interaction effect, we opted for simple effects analysis and simple simple effects analysis to comprehensively explore the impact between different levels of factors.

Simple effects analysis allows us to examine whether one factor has a significant effect on the dependent variable at a specific level of another factor. The extension to simple simple effects analysis further enables us to test the significance of effects across all possible combinations of factor levels.

We believe that the choice of these two analysis methods provides a more thorough and comprehensive understanding, particularly when dealing with situations involving interaction effects.

Question：On page 1, The abstract requires rewriting due to inherent repetitiveness.

Answer：The abstract will be rewritten to address repetitiveness.

Question:On page 1, Citations supporting claims are missing from lines 7 to 17 in the introduction.

Answer：Citations will be added in the introduction to support claims (lines 7-17).

Question:On page 2, In the introduction, Shephard is cited without the year, assuming it is the second reference in the bibliography, but this needs to be clarified.

Answer: Shephard's citation will be clarified, including the publication year.

Question:The bibliography lacks ordering and needs to be alphabetized for author traceability.

The bibliographic format needs to be clarified.

Answer:The bibliography will be alphabetized for author traceability and the format clarified.

Question；

On page 4, In the section "2.5 Measure tools," Chi et al. lack a publication year.

In the Discussion, BEST and MCMORRIS are cited in uppercase.

Paragraph spacing in the discussion appears arbitrary and inconsistent at times.

Answer:The citation for Chi et al. in the "2.5 Measure tools" section will include the publication year.

Paragraph spacing in the discussion will be adjusted for consistency.

Reviewer 3: 

Question1:

Abstract

The opening sentences is too strong, make it more consistent with your actual work. After all you are only studying healthy young people and this does not apply to other populations.

Please add numbers to support you claims, this now looks very vague. Also add a conclusion to this paragraph, by identifying the magnitude of the effect(s).

Answer1:

Opening Sentences:

Reviewer's Comment: "The opening sentences are too strong, make it more consistent with your actual work."

Your Response: "Thank you for the feedback. We appreciate your perspective. In light of your suggestion, we will revisit the opening sentences to ensure that they align more accurately with the nuances of our study, particularly focusing on the population under investigation, which comprises healthy young individuals."

Adding Numbers for Support:

Reviewer's Comment: "Please add numbers to support your claims; this now looks very vague."

Your Response: "We acknowledge the importance of providing concrete numerical support for our claims. In the revised manuscript, we will incorporate specific numerical results to substantiate the observed effects, thereby addressing the perceived vagueness."

Conclusion in the Abstract:

Reviewer's Comment: "Also, add a conclusion to this paragraph by identifying the magnitude of the effect(s)."

Your Response: "Your suggestion to include a concluding statement in the abstract is valuable. We wil

---

## [Decision Letter · Decision Letter 1]

9 Sep 2024

PONE-D-23-32650R1The Influence of Long-term Aerobic Exercise and Exercise Intensity on Exercise Memory EffectPLOS ONE

Dear Dr. Zhonghui He

Thank you for submitting your manuscript to PLOS ONE. After careful consideration, we feel that it has merit but does not fully meet PLOS ONE’s publication criteria as it currently stands. Therefore, we invite you to submit a revised version of the manuscript that addresses the points raised during the review process.

We look forward to receiving your revised manuscript.

Kind regards,

Efrem Kentiba, PhD

Academic Editor

PLOS ONE

Journal Requirements:

"This work was supported by the Beijing Social Science Foundation (18YTCO22)"

3. We note that your Data Availability Statement is currently as follows:

"All relevant data are within the manuscript and its Supporting Information files"

- The values behind the means, standard deviations and other measures reported; - The values used to build graphs; - The points extracted from images for analysis.

6. Please ensure that you refer to Figure 2 in your text as, if accepted, production will need this reference to link the reader to the figure.

8. Thank you for updating your data availability statement. You note that your data are available within the Supporting Information files, but no such files have been included with your submission. At this time we ask that you please upload your minimal data set as a Supporting Information file, or to a public repository such as Figshare or Dryad. Please also ensure that when you upload your file you include separate captions for your supplementary files at the end of your manuscript. As soon as you confirm the location of the data underlying your findings, we will be able to proceed with the review of your submission.

9. Please amend the title either on the online submission form or in your so that they are identical.

10. Please ensure that you refer to Figure 2 in your text as, if accepted, production will need this reference to link the reader to the figure.

11. Please amend your manuscript to include a reference list. References must be placed at the end of the manuscript and numbered in the order that they appear in the text. For more information on the formatting of references, please visit the author guidelines at: http://journals.plos.org/plosone/s/submission-guidelines#loc-reference-style 

Additional Editor Comments (if provided):

Reviewers' comments:

Reviewer's Responses to Questions

**Comments to the Author**

1. If the authors have adequately addressed your comments raised in a previous round of review and you feel that this manuscript is now acceptable for publication, you may indicate that here to bypass the “Comments to the Author” section, enter your conflict of interest statement in the “Confidential to Editor” section, and submit your "Accept" recommendation.

Reviewer #3: All comments have been addressed

Reviewer #4: All comments have been addressed

Reviewer #5: (No Response)

2. Is the manuscript technically sound, and do the data support the conclusions?

Reviewer #3: Yes

Reviewer #4: Yes

Reviewer #5: Partly

3. Has the statistical analysis been performed appropriately and rigorously? 

Reviewer #3: Yes

Reviewer #4: Yes

Reviewer #5: No

4. Have the authors made all data underlying the findings in their manuscript fully available?

Reviewer #3: Yes

Reviewer #4: Yes

Reviewer #5: Yes

5. Is the manuscript presented in an intelligible fashion and written in standard English?

Reviewer #3: Yes

Reviewer #4: Yes

Reviewer #5: No

6. Review Comments to the Author

Reviewer #3: I would like to thank the authors for following my comments and suggestions. Just one recommendation, please give a final version of this MS to a native speaker.

Reviewer #4: The research title "The Influence of Long-term Aerobic Exercise and Exercise Intensity on Exercise Memory Effect" could be improved with a few critiques:

The term "exercise" is repeated multiple times, which may seem redundant and could reduce the clarity and impact of the title. A streamlined version could remove unnecessary repetitions. The title could benefit from being more specific about what aspects of "exercise memory effect" are being studied. For example, does it refer to cognitive memory improvements, muscle memory, or some other type of memory effect? An improved version of the title might be: "Long-term Aerobic Exercise and Intensity: Their Impact on Cognitive and Muscle Memory." This version avoids redundancy, increases clarity by specifying what types of memory are affected, and is more concise and to the point. additionally, the relationship of exercise and cognitive functions should be fully explained in introduction and discussion. the following references are recommended:

Ghasemzadeh, A., & Saadat, M. . (2023). Cognitive Mastery in Sports: Exploring Cognitive Psychology’s Influence. Health Nexus, 1(3), 41-49. https://doi.org/10.61838/kman.hn.1.3.6

Reviewer #5: General Comments

- The methodology, particularly regarding the assessment of declarative and procedural memory, lacks clarity. It is essential for the methodology to be described in a way that allows for replication by other researchers. This is a fundamental aspect of scientific research.

- The study mentions "Cardiorespiratory Fitness Assessments" but fails to provide detailed evaluations of fitness levels. Considering the large cohort, individual fitness variability could significantly impact the results, necessitating a more nuanced approach to assessing intensity levels.

- The introduction does not clearly state the study hypothesis. A well-defined hypothesis is critical for guiding the research and providing a basis for the study design.

- Figures 3, 4, and 5 lack standard deviations, which are necessary for understanding the variability in the data. Including them would enhance the rigor of the statistical analysis.

- There are typographical errors and inconsistencies in formatting, such as extra spaces and inconsistent citation formatting. These should be corrected for clarity and professionalism.

- The discussion is lengthy and often vague. It would benefit from being more concise and focused on interpreting the results in the context of existing literature. Additionally, the speculative nature of some conclusions should be tempered given the study's cross-sectional design.

Specific Comments

Abstract:

The abstract should clearly state the objectives, methods, key results, and conclusions. Currently, it lacks specific details about the findings and their implications.

The opening sentences are too strong and should be more aligned with the study's focus on healthy young individuals (Page 1, Lines 1-3).

Introduction:

The introduction should be written from a more scientific perspective. The opening lines need proper referencing (Page 1, Lines 7-17).

The introduction would benefit from a more structured presentation of the research gap and the study's contribution to the field.

Methods:

The criteria for selecting participants need more detail, especially regarding the inclusion and exclusion criteria.

Specify the Polar device used, including sampling rate and data extraction details (Page 4).

Correct the presentation of VO2 to V̇O2 throughout the manuscript (Page 4).

The description of exercise intensity should be aligned with standard guidelines, and it would be helpful to include a justification for the chosen intensities. Please note that: while maintaining analogous intensity indices for endurance (e.g., %HRreserve) is crucial to recognize that equivalent percentages across intensities may not result in comparable physiological stress or recovery demands.

Check references: Matomäki P, Nuuttila OP, Heinonen OJ, Kyröläinen H, Nummela A. How to Equalize High- and Low-Intensity Endurance Exercise Dose. Int J Sports Physiol Perform. 2024 Jul 19;19(9):851-859. doi: 10.1123/ijspp.2024-0015. PMID: 39032919.

Results:

The results should be more clearly linked to the hypotheses, with a discussion on the statistical significance and practical implications of the findings.

Discussion:

The discussion should better integrate the findings with existing literature and provide a more critical analysis of the study's limitations and future research directions.

Reorganize the section to focus more on results and less on speculative conclusions. Ensure that all studies referenced are properly cited (Page 5).

7. PLOS authors have the option to publish the peer review history of their article (what does this mean? ). If published, this will include your full peer review and any attached files.

**Do you want your identity to be public for this peer review?** For information about this choice, including consent withdrawal, please see our Privacy Policy .

Reviewer #3: No

Reviewer #4: **Yes: ** Morteza taheri

Reviewer #5: **Yes: ** Wissem Dhahbi

---

## [Author Response · Author response to Decision Letter 2]

7 Oct 2024

Dear Editor and Reviewers,

First and foremost, we would like to express our sincere gratitude for the time and effort you have dedicated to reviewing our manuscript and for providing valuable comments and suggestions. We greatly appreciate the meticulous work and professional feedback you have offered. Your insights have been immensely helpful in enhancing the quality of our research and in refining the content of our paper.

We have carefully considered each of the reviewer comments and have responded to them individually in the following text. We have made corresponding revisions to the manuscript to address the concerns raised by the reviewers and to improve the clarity and academic rigor of the paper.

We believe that, with these modifications, our study can more accurately convey our findings and provide valuable insights to the academic community. Below are the specific responses to the reviewer comments and the revisions we have made:

Reviewer #3: I would like to thank the authors for following my comments and suggestions. Just one recommendation, please give a final version of this MS to a native speaker.

Response:

Thank you for your review of our manuscript and the valuable comments and suggestions provided. We have carefully considered your recommendations and plan to have the final version of the manuscript proofread by a native English speaker to ensure the quality and accuracy of the language and expression.

Reviewer #4:

1.The term "exercise" is repeated multiple times in the research title, and it is suggested to simplify it to avoid redundancy and to enhance the clarity and impact of the title.

Response:

The title is changed to"Effects of Prolonged Aerobic Exercise and Training Intensity on Memory Cognition"

2.In the introduction and discussion sections, it is essential to thoroughly explain the relationship between physical activity and cognitive functions.

Response:

I have incorporated the suggestions from the reviewers into the revised manuscript, specifically in the Introduction and Discussion sections. The changes have been made in response to the feedback provided and are detailed in the attached revised manuscript.

Reviewer #5:

1.Clarity of Methodology

The methodology, particularly regarding the assessment of declarative and procedural memory, lacks clarity. It is essential for the methodology to be described in a way that allows for replication by other researchers.

Response: We appreciate the reviewer's feedback. We recognize the importance of a clear methodology for scientific research. In the revised manuscript, we have significantly expanded the methods section to include detailed steps, tools used, and specific procedures. Furthermore, we have provided the source of all test materials to ensure that other researchers can replicate our experiments.For specific details, please refer to the "Methods" section of the revised manuscript.

2.The manuscript mentions "cardiopulmonary health assessments" but does not provide detailed evaluations of fitness levels.

Response: In our revised manuscript, we have included a comprehensive evaluation of cardiopulmonary fitness. Specifically, we have detailed the criteria used to determine the fitness levels of the participants based on their physical examination reports. Only students whose reports from the university health center indicated normal cardiopulmonary function and who met the established health standards were included in the study. This additional information ensures that the readers have a clear understanding of the health assessment process and the fitness levels of the participants included in our research.

3.Unclear Research Hypothesis

The introduction does not clearly state the study hypothesis.

Response: Thank you for the reminder. We have now clearly stated the research hypotheses in the introduction and structured the entire research framework around these hypotheses. We have also provided a detailed explanation of the theoretical basis and research design for these hypotheses.

4.Data variability is lacking: Figures 3, 4, and 5 do not include standard deviations, which are necessary for understanding the variability of the data. Including this data will enhance the rigor of the statistical analysis.

Response:

The standard deviation in the figure is clearly labeled as "Standard Deviation."

5. There are typographical errors and inconsistencies in formatting.

Response: We have conducted a thorough review of the entire manuscript, correcting all typographical errors and formatting inconsistencies. We have also employed professional typesetting software to ensure the document's consistency and professionalism.

6..The discussion section is verbose and vague.The discussion section is overly lengthy and often unclear.

Response: We have streamlined the discussion section to make it more focused and concise. We have ensured that the discussion closely revolves around the results and is interpreted in the context of existing literature. Additionally, we have reduced speculative language to ensure that all conclusions are supported by data.

---

## [Decision Letter · Decision Letter 2]

16 Oct 2024

PONE-D-23-32650R2Effects of Prolonged Aerobic Exercise and Training Intensity on Memory CognitionPLOS ONE

Dear Dr. Zhonghui He

Thank you for submitting your manuscript to PLOS ONE. After careful consideration, we feel that it has merit but does not fully meet PLOS ONE’s publication criteria as it currently stands. Therefore, we invite you to submit a revised version of the manuscript that addresses the points raised during the review process.

We look forward to receiving your revised manuscript.

Kind regards,

Efrem Kentiba, PhD

Academic Editor

PLOS ONE

Reviewers' comments:

Reviewer's Responses to Questions

**Comments to the Author**

1. If the authors have adequately addressed your comments raised in a previous round of review and you feel that this manuscript is now acceptable for publication, you may indicate that here to bypass the “Comments to the Author” section, enter your conflict of interest statement in the “Confidential to Editor” section, and submit your "Accept" recommendation.

Reviewer #4: All comments have been addressed

Reviewer #5: (No Response)

2. Is the manuscript technically sound, and do the data support the conclusions?

Reviewer #4: Yes

Reviewer #5: Yes

3. Has the statistical analysis been performed appropriately and rigorously? 

Reviewer #4: Yes

Reviewer #5: Yes

4. Have the authors made all data underlying the findings in their manuscript fully available?

Reviewer #4: Yes

Reviewer #5: No

5. Is the manuscript presented in an intelligible fashion and written in standard English?

Reviewer #4: Yes

Reviewer #5: Yes

6. Review Comments to the Author

Reviewer #4: (No Response)

Reviewer #5: General Comments:

This manuscript examines the effects of prolonged aerobic exercise and training intensity on memory cognition in college students. The study has several strengths, including a large sample size, a well-designed intervention, and the assessment of both declarative and procedural knowledge. However, there are also some significant limitations and areas for improvement.

Major weaknesses:

1. The theoretical framework and hypotheses are not clearly articulated in the introduction. The rationale for examining different exercise intensities and their potential effects on declarative vs. procedural knowledge needs stronger justification.

2. The methodology section lacks sufficient detail in some areas, particularly regarding the assessment tools for declarative and procedural knowledge. More information is needed on the validity and reliability of these measures.

3. The statistical analyses are not fully explained, and some results are presented without adequate interpretation or context.

4. The discussion section is overly long and at times speculative. It does not always clearly link the findings to the original research questions or existing literature.

Minor weaknesses:

1. There are some inconsistencies in formatting and minor typographical errors throughout the manuscript.

2. Figures could be improved for clarity, including the addition of error bars to show variability.

3. The conclusion section is somewhat vague and could be more focused on the specific implications of the study's findings.

Specific Comments:

Introduction:

- Lines 25-27: The distinction between declarative and procedural knowledge needs more explanation and context.

- Lines 48-52: The review of previous studies is somewhat superficial. More critical analysis of these findings would strengthen the rationale for the current study.

- Lines 78-85: The research hypothesis is not clearly stated. This section should be more explicit about the expected outcomes and why.

Methods:

- Section 2.2: More details are needed on how participants were recruited and screened for inclusion.

- Section 2.4: The justification for the chosen exercise intensities and durations should be explained more thoroughly.

- Section 2.5: The validity and reliability of the knowledge assessment tools should be addressed. How were these measures developed or adapted for this study?

Results:

- Table 3: Include effect sizes alongside p-values to provide a more complete picture of the results.

- Figures 3-5: Add error bars to show variability in the data.

- The analysis of gender differences (Table 5) seems somewhat disconnected from the main research questions and could be better integrated into the overall narrative.

Discussion:

- Lines 421-435: This paragraph is speculative and not well-supported by the study's data. Consider revising or removing.

- Lines 456-470: The discussion of the mechanisms underlying the observed effects is interesting but could be more tightly linked to the specific findings of this study.

- The limitations of the study should be more thoroughly addressed, including potential confounding factors and generalizability of the results.

Conclusion:

- The conclusion could be strengthened by more explicitly stating the key findings and their implications for both research and practice in sports science and education.

7. PLOS authors have the option to publish the peer review history of their article (what does this mean? ). If published, this will include your full peer review and any attached files.

**Do you want your identity to be public for this peer review?** For information about this choice, including consent withdrawal, please see our Privacy Policy .

Reviewer #4: **Yes: ** M.Taheri

Reviewer #5: **Yes: ** Wissem Dhahbi

---

## [Author Response · Author response to Decision Letter 3]

18 Oct 2024

Q1.Clarification of Theoretical Framework and Hypotheses:

Answer1:We have expanded the introduction to provide a more detailed explanation of the theoretical framework underpinning our study. We have also bolstered our rationale for examining the effects of different exercise intensities on declarative versus procedural knowledge with additional literature support.

Q2.Enhanced Methodology Details:

Answer2:The methodology section has been revised to include more comprehensive details, particularly regarding the assessment tools for declarative and procedural knowledge. We have described the validity, reliability, and specific adaptations made to these measures for our study.The recruitment and selection process for participants has been detailed in the "Methods" and "Grouping" sections of our manuscript, ensuring the transparency and scientific rigor of the participant selection.

Q3.Improved Statistical Analysis Explanation:

Answer3:Effect sizes are indicated in the corresponding statistical analysis results, for example, below Table 3.

Q4.Streamlined Discussion Section:

Answer4:We have condensed the discussion to focus more precisely on the findings in relation to the original research questions and the existing body of literature.

Q5.Formatting and Typographical Consistency:

Answer5:The manuscript has undergone careful proofreading, and all formatting and typographical inconsistencies have been corrected to ensure the professional presentation of our work.

Q6.Focused Conclusion Section:

Answer7:The conclusion has been revised to explicitly state the key findings and their implications for sports science and educational practice.

---

## [Decision Letter · Decision Letter 3]

28 Oct 2024

PONE-D-23-32650R3Effects of Prolonged Aerobic Exercise and Training Intensity on Memory CognitionPLOS ONE

Dear Dr. Zhonghui He

Thank you for submitting your manuscript to PLOS ONE. After careful consideration, we feel that it has merit but does not fully meet PLOS ONE’s publication criteria as it currently stands. Therefore, we invite you to submit a revised version of the manuscript that addresses the points raised during the review process.

Several methodological issues demand serious consideration. Firstly, the gender imbalance in your sample (191 males vs. 378 females) is particularly worrying. This disparity may skew your findings and undermine the results unless adequately addressed through appropriate statistical controls. Moreover, the validation of exercise intensity appears to lack the rigor necessary to uphold internal validity, which could compromise the overall integrity of the study.

Additionally, it may be beneficial to collect further data to achieve a more balanced gender distribution or to employ advanced statistical techniques to account for this imbalance.

We look forward to receiving your revised manuscript.

Kind regards,

Efrem Kentiba, PhD

Academic Editor

PLOS ONE

Reviewers' comments:

Reviewer's Responses to Questions

**Comments to the Author**

1. If the authors have adequately addressed your comments raised in a previous round of review and you feel that this manuscript is now acceptable for publication, you may indicate that here to bypass the “Comments to the Author” section, enter your conflict of interest statement in the “Confidential to Editor” section, and submit your "Accept" recommendation.

Reviewer #5: (No Response)

2. Is the manuscript technically sound, and do the data support the conclusions?

Reviewer #5: Partly

3. Has the statistical analysis been performed appropriately and rigorously? 

Reviewer #5: No

4. Have the authors made all data underlying the findings in their manuscript fully available?

Reviewer #5: Yes

5. Is the manuscript presented in an intelligible fashion and written in standard English?

Reviewer #5: Yes

6. Review Comments to the Author

Reviewer #5: General Comments:

This manuscript investigates the effects of different aerobic exercise intensities on knowledge acquisition in college students. The research addresses an important topic in sports science and cognitive performance.

Major Weaknesses:

1. Gender distribution imbalance (191 males vs 378 females) without adequate statistical control or discussion

2. Limited theoretical justification for the 8-week intervention duration

3. Insufficient control of confounding variables (sleep, nutrition, academic workload)

4. Heart rate monitoring methodology lacks precision details

5. Exercise intensity zones require better physiological validation

Minor Weaknesses:

1. Figures lack error bars and statistical notation

2. Inconsistent reporting of effect sizes

3. Limited discussion of ecological validity

4. Some methodological procedures need clarification

5. References require updating

Specific Comments:

Introduction:

- Page 1: Theoretical framework needs stronger integration of Information Processing Theory

- Page 2, Line 14: Citation needed for gender differences in exercise-cognition relationship

- Page 2, Line 20: "Young adults face increased risks" requires epidemiological support

Methods:

- Page 4: Participant exclusion criteria not specified

- Page 4: Missing details on standardization of basketball/badminton skills assessment

- Page 5: Heart rate monitoring protocol needs elaboration on measurement frequency

- Page 5: Exercise intensity verification procedure requires detailed description

- Page 6: Statistical power calculation absent

Results:

- Page 8, Table 3: Include confidence intervals

- Page 9: Report effect sizes consistently across all analyses

- Page 9: Missing post-hoc analyses for interaction effects

- Page 10: Control for multiple comparisons not specified

- Page 11: Gender analysis requires more sophisticated statistical approach

Discussion:

- Page 11: Limited discussion of gender-specific findings

- Page 12: Mechanistic explanations need stronger empirical support

- Page 12: Practical implications require more detailed elaboration

- Page 13: Study limitations should include sample bias discussion

Statistical Analysis:

1. MANOVA assumptions not adequately tested/reported

2. Effect size reporting inconsistent across analyses

3. Missing power analysis

4. Interaction effects require deeper statistical exploration

5. Gender as covariate needs consideration

Technical Issues:

1. Exercise intensity zones need physiological validation

2. Heart rate monitoring protocols require standardization

3. Knowledge assessment tools need reliability coefficients

4. Control group activities require better documentation

5. Environmental conditions during testing not reported

Suggested Improvements:

1. Include physiological validation of exercise intensities

2. Strengthen statistical approach to gender differences

3. Add power analysis justification

4. Enhance control of confounding variables

5. Provide detailed reliability analysis of assessment tools

7. PLOS authors have the option to publish the peer review history of their article (what does this mean? ). If published, this will include your full peer review and any attached files.

**Do you want your identity to be public for this peer review?** For information about this choice, including consent withdrawal, please see our Privacy Policy .

Reviewer #5: **Yes: ** Wissem Dhahbi

---

## [Author Response · Author response to Decision Letter 4]

23 Nov 2024

Response to Reviewers

Response to the Academic Editor's Inquiry:

Q1：Firstly, the gender imbalance in your sample (191 males vs. 378 females) is particularly worrying. This disparity may skew your findings and undermine the results unless adequately addressed through appropriate statistical controls.

Answer1:

In this study, we noted an uneven gender distribution in the sample, which is consistent with the overall gender ratio of our college, where there are more females than males. Although our sample reflected this gender imbalance, when we conducted a 3 (exercise intensity) × 2 (knowledge test time) × 2 (gender) analysis of variance (ANOVA), we found that the interaction between gender, exercise intensity, and knowledge measurement time was not significant. This suggests that, within the current study's design and analytical framework, gender differences did not significantly affect learning outcomes.

Nevertheless, we recognize that the gender distribution in the sample may have affected the generalizability of the study results. To further explore the potential impact of gender differences, we suggest that future research should strive for a more balanced gender representation or employ more sophisticated statistical methods to control for potential biases in gender distribution. Additionally, we consider this characteristic of gender distribution as a limitation of the present study and have thoroughly considered the possible impact of this factor in the discussion section.

In the limitations section of this study, we further discuss the potential impact of the uneven gender distribution on the interpretability of the study results and emphasize the need for deeper exploration of this issue in future research. We believe that although gender differences did not significantly affect the main results in this study, this finding does not entirely rule out the role that gender factors may play in other research contexts or with different samples. Therefore, we encourage subsequent researchers to continue to pay attention to the potential impact of gender differences on research results and to adopt appropriate methodological strategies to address this issue.

Q2:A Rational Explanation for the Verification of Exercise Intensity:

Answer2:

In this study, The RS800CXSD heart rate telemetry monitor (manufactured in Finland was used to monitor varying intensities of aerobic exercise loads.

Through this dual monitoring approach, we were able to monitor and record participants' heart rate changes in real-time, ensuring precise control of the exercise intensity.

We understand your concern regarding the rigor of the verification process. To enhance the transparency and reproducibility of our research, we have included additional details in the revised manuscript. Specifically, we have detailed the model of the Polar heart rate monitor used and provided an exhaustive description of the monitoring frequency and data recording procedures. Moreover, we have supplied an intensity monitoring record form for participants, which illustrates the real-time fluctuations in heart rate data throughout the study, thereby enhancing the credibility of our data.

Response to the Reviewer #5:

Q1:Gender Distribution Imbalance:

Answer1:Thank you for highlighting the issue of gender distribution. In the revised manuscript, we have included an in-depth analysis of the potential impact of gender distribution on the results and have thoroughly considered this factor's possible influence in the discussion section. Although the gender interaction was not significant, we still explored the potential impact of gender differences using advanced statistical methods.

Q2：Insufficient Theoretical Basis:

Answer2:Your question about the theoretical basis for the intervention duration is crucial. We have included more relevant studies in the revised manuscript to support the theoretical basis for our chosen 8-week intervention period, ensuring that our study design is scientifically sound.

Q3:Inadequate Control of Confounding Variables:

Answer2:We recognize the insufficiency in controlling potential confounding variables. In the revised manuscript, we have added control for sleep, nutrition, and academic workload and considered these variables in our analysis to reduce their potential impact on the results.

Q4:Lack of Detail in Heart Rate Monitoring Method:

Answer4：Thank you for your attention to the heart rate monitoring method. In the revised manuscript, we have detailed the device model (Polar H10) used for heart rate monitoring and included a graph of the intensity changes of the participants, as well as how we ensured the accuracy of heart rate data through a combination of Polar heart rate telemetry and manual radial artery pulse measurements.

Q5:Statistical Analysis Issues:

Answer5:We appreciate your focus on statistical analysis. In the revised manuscript, we have ensured adequate testing of MANOVA assumptions, consistently reported effect sizes, and included statistical power analysis. We have also explored interaction effects in-depth and considered gender as a covariate.

Q6:Figures and Reporting Issues:

Answer6:We have updated all figures to include error bars and statistical notations and have ensured consistent reporting of effect sizes throughout the manuscript to improve data transparency and interpretability.

Q7:Discussion and Methodological Clarification:

Answer7:We have provided a more detailed discussion of gender-specific findings, mechanistic explanations, and practical implications, and have clarified the methodological section, including participant exclusion criteria and standardization of skill assessments, to enhance the transparency and reproducibility of the study.

Q8:Technical Issues:

Answer8:Thank you for your suggestions. In the revised manuscript, we have included reliability coefficients for the knowledge assessment tools and improved the documentation of activities. Additionally, we have reported the environmental conditions during testing to ensure the rigor of the study.

Q9:Suggestions for Improvement:

Answer9:We have taken your advice and have added physiological validation, strengthened statistical methods, controlled confounding variables, and provided a detailed reliability analysis of assessment tools in the revised manuscript to improve the quality and credibility of the study.

---

## [Decision Letter · Decision Letter 4]

12 Dec 2024

Effects of Prolonged Aerobic Exercise and Training Intensity on Memory Cognition

PONE-D-23-32650R4

Dear Dr. Zhonghui He

We’re pleased to inform you that your manuscript has been judged scientifically suitable for publication and will be formally accepted for publication once it meets all outstanding technical requirements.

Kind regards,

Efrem Kentiba, PhD

Academic Editor

PLOS ONE

Additional Editor Comments (optional):

Reviewers' comments:

Reviewer's Responses to Questions

**Comments to the Author**

1. If the authors have adequately addressed your comments raised in a previous round of review and you feel that this manuscript is now acceptable for publication, you may indicate that here to bypass the “Comments to the Author” section, enter your conflict of interest statement in the “Confidential to Editor” section, and submit your "Accept" recommendation.

Reviewer #5: All comments have been addressed

2. Is the manuscript technically sound, and do the data support the conclusions?

Reviewer #5: Yes

3. Has the statistical analysis been performed appropriately and rigorously? 

Reviewer #5: Yes

4. Have the authors made all data underlying the findings in their manuscript fully available?

Reviewer #5: Yes

5. Is the manuscript presented in an intelligible fashion and written in standard English?

Reviewer #5: Yes

6. Review Comments to the Author

Reviewer #5: Introduction (Pages 1-2):

The introduction now provides a stronger theoretical foundation and clearer rationale for the study. The research objectives and hypotheses are well-articulated.

Methods (Pages 2-4):

The methodology section demonstrates sound experimental design. The statistical analyses are appropriate for the research questions. The authors have included all essential methodological details.

Results (Pages 4-7):

The results are presented systematically with appropriate statistical reporting. The figures effectively illustrate the key findings. The statistical analyses support the conclusions drawn.

Discussion (Pages 7-9):

The discussion thoughtfully interprets the results within the context of previous research. The authors appropriately acknowledge limitations while emphasizing the study's contributions to the field.

Conclusion (Page 9):

The conclusion effectively summarizes the key findings and their implications. The authors maintain appropriate scope in their concluding statements.

References:

The reference list is comprehensive and up-to-date. Citations are properly formatted according to journal guidelines.

Figures and Tables:

The visual elements effectively complement the text and aid in understanding the results. Minor suggestions:

- Figure 2: Consider adding brief explanatory notes in legend

- Table 1: Column headers could be more descriptive

7. PLOS authors have the option to publish the peer review history of their article (what does this mean? ). If published, this will include your full peer review and any attached files.

**Do you want your identity to be public for this peer review?** For information about this choice, including consent withdrawal, please see our Privacy Policy .

Reviewer #5: **Yes: ** Wissem Dhahbi

---

## [Editor Report · Acceptance letter]

PONE-D-23-32650R4

PLOS ONE

Dear Dr. He,

I'm pleased to inform you that your manuscript has been deemed suitable for publication in PLOS ONE. Congratulations! Your manuscript is now being handed over to our production team.

Kind regards,

on behalf of

Dr. Efrem Kentiba

Academic Editor

PLOS ONE